# The feasibility analysis of integrating community-based health insurance schemes into the national health insurance scheme in Uganda

Prossy Kiddu Namyalo[1][*], Boniface Mutatina[2], Sarah Byakika[3], Aliyi Walimbwa[3], Rose Kato[4], Robert K. Basaza[5]

1 Faculty of Social Sciences, Ndejje University, Kampala, Uganda, 2 School of Medicine, Makerere University, Kampala, Uganda, 3 Health Services, Planning, Financing and Policy Department, Ministry of Health, Kampala, Uganda, 4 Belgian Development Agency (Enabel), Kampala, Uganda, 5 School of Public Health, Gudie University Project, Kampala, Uganda

◉ These authors contributed equally to this work.
* pnkiddu@gmail.com

**Data Availability Statement:** All relevant data are within the manuscript and its Supporting Information files.

## Abstract

### Background

Uganda has a draft National Health Insurance Bill for the establishment of a National Health Insurance Scheme (NHIS). The proposed health insurance scheme is to pool resources, where the rich will subsidize the treatment of the poor, the healthy will subsidize the treatment of the sick, and the young will subsidize the treatment of the elderly. However, there is still a lack of evidence on how the existing community-based health insurance schemes (CBHIS) can fit within the proposed national scheme. Thus, this study aimed at determining the feasibility of integrating the existing community-based health financing schemes into the proposed National Health Insurance Scheme.

### Methods

In this study, we utilized a multiple–case study design involving mixed methods. The cases (i.e., units of analysis) were defined as the operations, functionality, and sustainability of the three typologies of community-based insurance schemes: provider-managed, community-managed, and third party-managed. The study combined various data collection methods, including interviews, survey desk review of documents, observation, and archives.

### Findings

The CBHIS in Uganda are fragmented with limited coverage. Only 28 schemes existed, which covered a total of 155,057 beneficiaries with an average of 5,538 per scheme. The CBHIS existed in 33 out of 146 districts in Uganda. The average contribution per capita was estimated at Uganda Shillings (UGX) 75,215 = equivalent to United States Dollar (USD) 20.3, accounting for 37% of the national total health expenditure per capita UGX 51.00 = at 2016 prices. Membership was open to everyone irrespective of socio-demographic status.

**Funding:** The authors received no specific funding for this work.

**Competing interests:** The authors have declared that no competing interests exist.

**Abbreviations:** CBHI, Community Based Health Insurance; CBHIS, Community Based Health Insurance Schemes; CHI, Community Health Insurance; CSOs, Civil Society Organisations; DHIS, District Health Information System; DHIS, District Health Insurance Schemes; DHO, District Health Officer; DMHIS, District Mutual Health Insurance Scheme; DMHIS, District Mutual Health Insurance Schemes; HIV, Human Immuno Deficiency Syndrome; HSDP, Health Sector Development Plan; IPD, In-Patient Department; KI, Key Informant; KIIs, Key Informant Interviews; MHOs, Mutual Health Organisations; MHOs, Mutual Health Organizations; MOH, Ministry of Health; MOU, Memorandum of Association; NHIA, National Health Insurance Authority; NHIF, National Health Insurance Fund; NHIS, National Health Insurance Scheme; OPD, Out-Patient Department; RSSB, Rwanda Social Security Board; SACCO, Savings and Credit Cooperatives Organizations; UCBHFA, Uganda Community Based Health Financing Association; UHC, Universal Healthcare Coverage; USD, United States Dollar; WHO, World Health Organization.

The schemes had inadequate capacity for management, strategic planning, and finances and lacked reserves and reinsurance. The CBHIS structures included promoters, the scheme core, and the community grass-root structures.

## Conclusion

The results demonstrate the possibility and provide a pathway to integrating CBHIS into the proposed NHIS. We however recommend implementation in a phased manner including first providing technical assistance to the existing CBHIS at the district level to address the critical capacity gaps. This would be followed by integrating all three elements of CBHIS structures. The last phase would then involve establishing a single fund for both the formal and informal sectors managed at the national level.

## Introduction

Financial risk protection is a key element of universal health coverage (UHC), along with access to quality essential healthcare and access to safe, effective, quality, and affordable essential medicines and vaccines for all [1]. Developing countries are increasingly expanding national health insurance to achieve their universal health coverage targets by 2030. By providing protection against catastrophic health expenditure among insured households, health insurance improves access and utilization of good quality health services [1–4]. In some sub-Saharan African countries such as Ethiopia, efforts have focused on implementing community-based health insurance schemes (CBHIS) on large scale, where communities are actively engaged in resource mobilization and management [4].

Like some other developing countries, Uganda is in the process of establishing a national health insurance scheme to offer financial risk protection to its citizens where catastrophic health expenditure is currently beyond the set threshold of 10% and 25% [5]. A National Health Insurance Scheme (NHIS) Bill, 2019 is pending reintroduction to the 11th Parliament for fresh discussions after it was called [6, 7]. The broad objective of the NHIS Bill is to provide for the establishment of the NHIS. Among other things, the Bill provides for the registration of contributors and their dependents, accreditation of health service providers, and articulation of benefits available to members under the Scheme. The bill also provides for the NHIS to determine and register persons who qualify as vulnerable groups or indigents—those that cannot afford to contribute to NHIS—to equally have access to the benefits specified for members of the scheme [6]. The effort towards the creation of a national scheme comes at a time when there are already existing CBHIS. The existence of these schemes can be traced as far back as 1995; when the first scheme was piloted at Kisiizi Hospital with support from the Department for International Development of the United Kingdom and the government of Uganda [8]. Uganda's community-based health insurance schemes have since formed an umbrella association with 25 members [8].

Despite the existence of several CBHIS in the country, there is a lack of evidence on how best to integrate them into the proposed NHIS. The CBHIS were not fully studied to understand their financial and sustainability capacity, functionality, and current practices. It was not clear what process would be followed to integrate the schemes. The integration of CBHIS into NHIS often requires stewardship, legislation, government political commitment, regional-level professional management, funding support, and continuing community-level support

[9]. As such, this study sought to analyze and document Uganda's operations, functionality, and sustainability of community-based health insurance schemes. Specifically, we were interested in establishing the national coverage, the average contribution per capita and the number of beneficiaries per scheme, and the percent of total health expenditure managed under the Community-based health insurance schemes. We also wanted to understand the financial health of the schemes and their current practices in terms of pooling mechanism, member enrollment, and retention. From the onset, we were interested in providing a comprehensive account of CBHIS in Uganda and determining the feasibility of integrating them into the NHIS. Thus, this paper demonstrates the possibility and provides the pathway for integrating CBHIS into the proposed NHIS.

## Methods

### Study design

A multiple-case study design involving mixed methods was utilized. The multiple-case study design was selected because it provided an opportunity to explore in-depth community-based health insurance schemes. The cases (i.e., the unit of analysis) were defined as the operations, functionality, and sustainability of the three typologies of community-based insurance schemes: provider-managed, community-managed, and third-party-managed [8, 10]. Furthermore, the multiple-case study design provided a better understanding of the three scheme types and how best they can be an integral part of the national health insurance scheme. It also provided an opportunity for the selection of several instrumental bounded cases to develop a more in-depth understanding of the phenomena than a single case can provide. This study combined various data collection methods, including interviews, a survey, and a desk review of documents.

### Study setting

Data were collected from seven (7) districts across the country. A district is a type of administrative unit in Uganda, managed by the local government. Uganda has no regional Administration. We purposively selected districts that were covered by at least one of the three CBHIS typologies. The districts of Bushenyi, Rukiga, and Rukungiri were considered for provider-managed schemes. While Mubende and Nakaseke districts were considered for the community-managed schemes. Lastly, Luwero and Masaka districts were selected for the third-party managed schemes. The study districts, CBHIS types, and service providers are provided in S1 Table.

### Data collection

The study team reviewed various documents on the operation of CBHIS in Uganda using a specially developed analytical guide. The reviewed documents included: CBHIS reports and strategic plans; NHIS Bill, Health Sector Development Plan (HSDP), 2015–2020; and Second National Development Plan; HSDP Mid-term Review. Others were the National Health Accounts, Annual Sector Performance, Health Financing Strategy and districts performance reports, and Uganda Community Based Health Financing Association reports. The study team visited offices and searched websites of the relevant government institutions, Civil Society Organisations (such as the Uganda Community Based Health Financing Association and Save for Health Uganda), and other key stakeholders to obtain the relevant documents. In addition, the team also reviewed global literature including peer-reviewed articles on CBHIS, World Health Organisation (WHO) guidelines and publications on CBHIS, International Labour

Organization publications on health microinsurance schemes, Country Acts, reports on NHIS and National Health Insurance Funds (NHIF), and published literature from Ghana, Rwanda, and Tanzania where integration of CBHIS has been undertaken. Tanzania and Rwanda were selected on basis of their geographical location in the East African Region where Uganda lies while Ghana was selected because of her similar development history to Uganda since its independence from the colonial master. These countries have also made positive steps in integrating CBHIS into NHIS. The key questions addressed in the literature review included: What was the objective of the integration? How did the issue of integration come to the table (Agenda Setting) and what are the key processes or stages of integration? How were the stakeholders managed? How long did the process take? What is the status and practice? What are the current challenges and how are they being addressed? The team used the Google search engine to search for peer-reviewed articles published in English and checked the reference lists of the documents found to expand the list of the included literature. The CBHIS and Uganda Community-Based Health Financing Association (UCBHFA) records were also reviewed to capture specific scheme information.

A total of 33 key informant interviews (KIs) were conducted using a specially developed interview guide to obtain relevant information and elicit respondents' reactions on CBHIS functionality, financial performance and sustainability, best practices, and their integration into NHIS. The key informant guide was administered to policymakers and implementers both at the district and national levels, scheme leadership, and members of civil society. A list of interviewed KIs is provided in S2 Table. Also, a survey of CBHIS was carried out using a capacity assessment tool that was administered to the scheme managers to examine the functionality of CBHIS in terms of six dimensions: (i) governance and legal structures (for example whether the CBHIS has a functional board and policy documents); (ii) financial management and internal control structures; (iii) administration systems; and strategic planning and sustainability; (iv) financial reserves (v) re-insurance practice (vi) guarantors. The capacity assessment findings are presented in S8 Table. Written informed consent was obtained from all participants for inclusion in the study.

## Data management

Data collected from the capacity assessment tool, interviews, and syntheses of the documents were thematically analyzed. The audio recordings from interviews were first transcribed. The analysis was done in a phased manner, beginning with reading and re-reading the transcriptions to become familiar with what the data entails; followed by the formation of comprehensive codes. Data were collapsed into codes for more efficient analysis. To enhance the rigor of the analysis, more than one qualitative expert regularly consulted with each other to generate consensus on the codes. Codes were then combined into over-arching themes that accurately depict the data to list candidate themes for further analysis. The team looked at how the themes supported the data and the over-arching perspectives to have coherent recognition of how the themes were patterned to present an accurate synthesis of the data. The insights drawn from the qualitative data analysis enabled the team to develop comprehensive accounts of the different aspects of CBHIS functionality, financial performance, sustainability, best practices, poor performances, and integration into NHIS.

Quantitative data obtained from UCBHFA, CBHIS records, and documents review were cleaned and manually edited for any irregularities. The data were entered on screens designed using Microsoft Excel. The data captured in Microsoft Excel were exported to STATA for more detailed analysis. Logical checks and frequency runs were made on all variables to improve the accuracy and consistency of the data and identify any outliers before actual data

analysis. The data were analyzed in line with the study objectives. Two validation meetings were held with key stakeholders (24) to authenticate and make inputs in draft reports. This manuscript was drafted from a study that was commissioned by the Ugandan ministry of health whose report is available on its website.

### The scientific advisory committee and ethical committee

A team of experts from the Uganda Social Protection Unit of the Ministry for Gender, Labour and Social Development, United States Development Aid, Belgian Development Cooperation, and the Ministry of Health of Uganda constantly securitized the study design, results, and reports of this research. They made useful comments which shaped the study. In addition, the results of the study were presented at various fora of the Ministry of Health headquarters and provided useful input into the study report.

### Ethical consideration

The study was approved by Clarke International University Research and Ethics Committee (No: Clarke-2020-10). Written informed consent was obtained from all participants for inclusion in the study. Participants were assured that the information provided would be kept securely and only accessed by the study team and will only be used for the purpose of this study. The Ministry of Health provided a letter of introduction to respondents. The participants were informed that the information they provided could assist the health sector in integrating the existing Community-Based Health Insurance Schemes into the proposed National Health Insurance Scheme. The participants were also informed of their right to walk out of the interview if they wished to withdraw.

## Results

### Experience in integration processes from other sub-Saharan African countries

Rwanda and Ghana provided good examples in sub-Saharan Africa where the integration of CBHIS into national health insurance schemes has been successful. In Rwanda, the integration process is reported to have significantly increased outpatient service utilization rates. The country has one national coordination mechanism that manages over 500 pools at the community level, 30 district pools, and 1 national pool. One of the key best practices in Rwanda is the separation of responsibilities where the Ministry of Health is responsible for policy formulation and service provision at health facilities and hospitals and the Rwanda Social Security Board (RSSB) is charged with policy implementation and purchasing [11–17].

In Ghana, before integration, over 145 District Mutual Health Insurance Schemes (DMHIS) and other voluntary Mutual Health Organisations (MHOs) operated autonomously with just oversight from the National Health Insurance Council. In 2012, the then-new Parliamentary Act saw the disjointed DMHIS get consolidated into a single national health insurance system under the management of the National Health Insurance Authority (NHIA). At the time of this study, the scheme had one governance and administrative system with powers to accredit service providers. The individual MHOs and their networks were first forced to suspend their operations or merge with the district schemes. The NHIA licenses and regulates the DMHIS [18–25].

In both Ghana and Rwanda cases, integration was done in a phased manner. The fragmented community-based schemes were first consolidated into district-level schemes under the supervision of a government body. Later the district-level schemes became strengthened

and integrated into the national insurance schemes [11, 13, 14, 20–22]. A synthesis of the literature review from the three countries is presented in S3 Table [11–30]. Whereas integration has not been successful in other countries like Tanzania [26–30], Uganda can still draw some lessons from its experience. For instance, Uganda can learn from Tanzania's experience where integration was hampered by politics and the fact that the idea was being pushed by development partners with the government taking a back seat.

## The functionality of community-based health insurance schemes in Uganda

The functionality of CBHIS was assessed in terms of the national coverage and the average number of beneficiaries per scheme, governance and management structures, enrolment of members, criteria for determining contributions, and the corresponding benefit packages. Other additional functionality aspects considered were timing and terms of contributions, availability of subsidies and grants to CBHIS, accessibility of the CBHIS by the different population groups, and performance of memoranda of understanding.

### CBHI national geographical coverage and current enrolment

At the time of this study, there were 28 CBHI schemes in Uganda getting services from 65 healthcare facilities. These schemes were located in a few districts (n = 33 districts, 22%) throughout the country with the largest number in Southwestern Uganda (S4 Table). Some of these were implemented under unions or networks. There were 3 types of schemes: provider, community, and third-party managed. The majority of the CBHIS were provider-managed (n = 21, 75%), followed by community-managed (n = 4, 14%), and a few third-party-managed (n = 2, 7%). One scheme was transitioning from being a community-managed scheme to a provider-managed one. As of 2019, a total of 155,057 beneficiaries were covered by the 28 CBHIS with an average enrolment of 5,538 beneficiaries per scheme. The scheme with the least number of beneficiaries had 64 while the highest had 43,615.

### Penetration rate and other CBHIS performance indicators

The community-based health insurance schemes in Uganda cover 7.6% of their target population. In addition, the study team found out that CBHIS meets up to 37% of the total health expenditure and contributes about 27% of the annual budgets of the facilities that provide services. Thus, CBHIS contribute significantly to the health sector where they exist. During the interviews, one of the health facility administrators was very grateful for the income brought in through the CBHIS:

> "………I think the CBHIS finances about 20% especially when they pay in time. They indeed have a significant contribution to the health centre's annual budget" (Health Facility Administrator, 1).

Another finding showed that the average contribution per capita (member) for CBHIS was Uganda Shillings (UGX) 75,215 = (USD 20.3) (Midyear exchange rate 2021, 1USD = 3701 UGX). It was noted that families are quick to buy cheaper benefit packages. This contribution seems huge for an average Ugandan to pay for their family members. A recent estimate in 2019 showed that 21.4% of Ugandans are below the poverty line of $1.90 a day [37]. Thus, the low penetration rate may be attributed to the fact that people cannot afford the premiums. One Scheme Manager noted that:

*"…….. majority of the community are poor and cannot afford healthcare and as such don't join CBHIS". Even with low premiums, some families will not afford to register with the scheme" (Scheme Manager, 6).*

The proportion of the premium used for administrative costs was found to be 14.2%. This implies that the biggest portion of the premiums is put into the healthcare fund. The administrative overhead costs were highly subsidized especially in the community and the third-party managed compared to the provider managed. The subsidies were from the promoting institution sometimes through donor funding or health facility finances.

## Criteria for determining contribution and benefit package

The findings show that in community and third-party managed schemes, the promoters of the CBHIS were doing some form of a feasibility study. They would identify the cost of care from potential service providers and later ask the people what they would like to have as the benefits package. On the other hand, the provider-managed schemes did not conduct feasibility studies because they felt that they have the required information. This information helps in designing and pricing benefit packages. Promoters then market and sell the designed packages within the target communities. The schemes' benefits packages excluded cosmetics care and self-inflicted injuries among others. Some schemes' benefits packages excluded chronic illnesses, while in other instances the promoters designed a special, expensive package for such conditions.

In response to how the scheme determines contribution and benefits package, some of the scheme managers explained that:

*"The beneficiaries decide about the services they want, we always seek for their feedback, but we technically compute and advise them accordingly" (Scheme Manager, 1).*

*"……...we develop several products catering for different services and needs, then calculate a premium for each as explained above. We sell all of them and the family makes a choice. We have exclusions and care for chronic diseases is not covered in the CBHI. We only cover both in-patient department and out-patient department care (IPD and OPD) and minor accidents. The members during a general meeting determine the services" (Scheme Manager, 2).*

From the beneficiary's point of view, it was noted that people are quick to buy a cheaper package:

*"We in the community decide on the package based on how much we pay. Inclusions and exclusions also depend on the fund we have collected but generally, the scheme offers all services available at a particular facility" (Scheme Leader/Beneficiary, 4).*

It was revealed that all packages irrespective of the type of the scheme have a co-payment embedded in them. The co-payment ranges from UGX 3,000 = (USD 0.8) to UGX 20,000 = (USD 5.4). This depends on the type of service and the cost of the package. The cost of co-payment is normally determined at the time of costing the benefits package. It was reported that the co-payment is intended to control abuse of the healthcare fund, reduce the frequency, and ensure that funds do not run out before the required period.

The packages have a ceiling that varies across the schemes. In all the CBHIS assessed, the ceilings are set on the amount of money one can get from the health fund per episode. However, in the third-party managed schemes on some packages, the ceiling was set on the

episodes per beneficiary in the family. Details of different packages and ceilings from third-party managed schemes are provided in S5 Table.

**Enrolment of beneficiaries.**   Regarding member enrolment, the study team found out that there was no harmonized way of enrolling members across the schemes; even within the same type of CBHIS. However, it was established that the unit of enrolment was family (household) and group. In this context, a family is defined as a group of people who eat from the same pot [10]. Individual enrolment was not encouraged except if someone was living alone. Individuals were also allowed to join CBHIS which uses Savings and Credit Cooperatives Organizations (SACCO) to enroll members such as Munno Mu Bulwadde Cooperation Mechanism. Enrolment is voluntary in all types of schemes, and it's done quarterly, bi-annually, or annually.

Irrespective of scheme type, the promoters first sensitize the target community about general health insurance. The target audience would be informed about the different packages when they agree to join. It was noted that the premium payments and amounts differ across the schemes; the situation was not different even within the same type of schemes. For example, within the provider-managed schemes, some schemes pay quarterly, bi-annually, and annually. The premium payment is charged per person but sometimes, a family premium is designed. Also, to encourage families not to hide some members (adverse selection), they try to reduce the premium of say the 5th plus the additional members. Enrolment was different where a CBHI targeted groups. In a group, families and their members still pay the premiums. However, within groups, its members were allowed to start accessing healthcare if they paid at least 60% to 80% of the expected premiums of all the families and its members. At this time, the families are issued the healthcare access identification card as they pay the balance. It was noted that premiums depended on the existing cost of care. However, the increase is capped at UGX 1,000 = (USD 0.3) and it does not occur every year. Sometimes a CBHI would spend two to three years without changing the premium.

**Timing and terms of contributions.**   There were differences in the timing of payment across the schemes. The enrolment approach determined the timing for premiums. It was noted that those collecting the premiums on an annual basis, would target to collect premiums during the September to December season. The in-kind payments were not allowed in any type of CBHI because of reported difficulties in computing equivalents in cash. It was also noted that all schemes accept payment in installments.

**Accessibility of CBHIS across the different population groups.**   The study aimed to explore the accessibility of the CBHIS across different population groups, such as persons with disability, women, children, and other vulnerable populations. The findings showed that all types of CBHI allow everyone to join irrespective of socio-demographic status. Despite this, it was reported that vulnerable and poor people were often left out because they couldn't afford to pay the premium. Also, the children were not allowed to join independently. It was noted that even though the schemes allowed all families to join; almost all schemes never keep the socio-demographic records. For example, it was difficult to ascertain the actual number of beneficiaries with disability across the board.

**Performance of Memorandum of Understanding.**   All CBHIS had a formal Memorandum of Understanding (MOU) with service providers. With exception of one scheme, which had delayed paying the provider (had not paid and at the same time did not communicate), other schemes and the service providers were satisfied with the performance of the MOU.

One of the administers for health facility noted that:

*"The MOU agreement highlights terms of services including the price. Then after the facility starts to offer the services. . .. . .they pay based on the service contract and MOU that we have*

*with them. We have never had any problems with them until this year when they delayed paying for 3 months without any communication. But they later paid . . . . . ." (Health Facility Administer, 1).*

All Memoranda of Understanding were reviewed on an annual basis. It was reported that some beneficiaries complain about the schemes when the terms in MOU are not respected.

*". . . ..we come here and directly complain. One time I came here, and they billed me wrongly, so I refused to pay. As a leader, if a member gets a problem while here, they phone me, and I came over to a meeting with administrators. We agreed that the facility sends us a monthly invoice within 14 days after the end of the month, so if they do, we scrutinize the bills and pay through the bank." (Scheme Leader, 2).*

## Financial sustainability of CBHIS in Uganda

This was assessed by examining the functionality of CBHIS' governance and legal structures; financial management and internal control structures; and whether they were doing strategic planning. The study team also assessed how CBHIS practiced risk-pooling and strategic purchasing. These aspects are summarised and categorized by the different types of CBHIS in S6 Table.

**Functionality of CBHIS governance, management, and legal structures.** The functionality of the governance structures of CBHIS was assessed in terms of the Board composition and responsibility. The study team assessed whether the external boards of CBHIS were comprised of members drawn from a broad spectrum and were independent of management. The study team also assessed whether the board organized regular and documented meetings with quorum.

Findings from the assessment indicate that generally, the schemes had functional external boards that were comprised of members drawn from a broad spectrum. To a larger extent, the boards were reported to be independent of management. However, among the third party-managed schemes, the independence of the board from management was very limited. It was also reported that in all CBHIS, the boards often held meetings.

Regarding the management structures, the provider-managed CBHIS generally had well-documented organizational structures that were being followed. Both the community and third-party managed schemes had well-documented organizational structures. However, some need to be updated. The organizational structures of the third-party managed schemes were not usually followed.

The study team found that due to financial constraints, both the governance and management structures of the Uganda Community-Based Health Financing Association, at the time of this study, had limited functional capacity. Between March 2014 and 2017, the secretariat was manned by one volunteer staff, and since then, it has been none functional despite the existence of a filled association's governance structure. The current board (which has 9 members) last held meetings in 2015 and the general assembly was last held in 2013. The organization has a constitution but it's not being implemented because there are no funds to run the activities.

When asked about the role of the association, one of the key informants narrated:

*". . . . . .of course, we miss the UCBHFA platform in bringing together the schemes, and right now we are lost like each promoter is doing what they want to do and there is no mechanism for bringing us together. However, UCBHFA was playing a big role and was a platform for*

*regulation, a voice for us as schemes, and giving CBHIS technical support" (Scheme Promoter, 2).*

On legal structures, only one out of the seven schemes was fully registered with the district as a Community-Based Organization. The other schemes lacked explicit legal registration and standalone entities. Of these, one scheme had applied for legal status but had not yet been granted at the time of the study. The study found out all seven schemes had constitutions that were amended.

**Financial management and internal control structures.** The CBHIS were not managed by registered accountants on a day-to-day basis. However, the schemes relied on accountants from the promoting organizations or the health facility. The study team also found that in the third-party and community-managed CBHIS, the staff of the promoters always audit the CBHIS on an annual basis. Regarding provider-managed schemes, the auditing was part of general health facility auditing and this is mainly external. In the same vein, the provider-managed budgets are incorporated into the health facility's annual budget.

**Strategic planning and sustainability.** The study assessed the existence of reserves, guarantors and re-insurance plans, pooling, and purchasing practices. Each of the provider-managed schemes was found to have a provision for a reserve fund, although more than three-quarters of them were empty.

Only one provider-managed scheme had a guarantor, which was the service provider itself. In this case, it was discovered that when the scheme's healthcare and reserve funds are depleted before the required period, the facility continues to provide the same services on credit until the expiry of the period. The scheme is then made to pay the debt in the subsequent period of premium payment.

The other schemes depend on the general assembly's resolution as noted by one of the scheme managers:

*"The scheme has a reserve account. However, in case it happens the general assembly and Board of Directors can decide to lobby for other funding from members to top up. It is the responsibility of the secretariat to ensure it does not happen for they are indicators and alert the board; in any case, services must continue to be offered" (Scheme Manager, 3)*

All the schemes lacked a re-insurance policy. However, most of the schemes were pooling the funds into one healthcare fund irrespective of the benefit packages.

*"It happened to one group, and we asked them to pay a premium top up. We have reserve fund but no re-insurance" (Scheme Manager, 5).*

All CBHIS employed a Scheme Manager. Part of the work of this manager is to monitor the healthcare fund utilization on a daily and or monthly basis. Despite this, it was reported that all CBHIS always had their healthcare funds depleted before the required period would end and they rely on reserve funds to enable beneficiaries to complete the coverage period. The depletion was mainly attributed to: (i) the inclusion of mainly the sick; (ii) the increasing cost of healthcare; (iii) and not charging actual premiums due to affordability challenges among other issues.

**The practice of risk-pooling and subsides to CBHIS.** Regardless of the unit of enrolment, the amount charged, and the benefits package, all the schemes maintain one pool for all their beneficiaries. This is evidence of both vertical and horizontal subsidization among CBHIS. This was confirmed by several respondents:

*"We have a single pool for all groups" (Scheme Manager, KI-7).*

*".……….we do pool or network the healthcare scheme fund. We have small schemes at the parish level. But, in every region, these come together in a network, and in doing so, they put together all their money into one pool on one account.…….. So, this helps to build a bigger pool that is able to pay the healthcare bills of their members without caring much about which scheme has contributed how much. Today, we are able to track and see who is consuming what amount and how much" (Scheme Promoter, KI 17).*

Some contracted providers subsidized schemes. Kiwoko hospital gives a 10% deduction on all bills of healthcare from scheme members. It was so far, the only facility that does this.

*".... they give us a discount on each bill when we are paying. So, we don't pay 100% of the issued bill but 10% less. We deduct it at the time of payment. The scheme managers calculate it. It was initially 5% but it has kept increasing over the years" (Scheme Leader/Beneficiary, KI 11).*

Across the schemes, it was reported that promoters support the schemes with some start-up funds. The study also established that the schemes are heavily subsidized by their promoters, especially for administration costs. Apart from one scheme in the Southwestern, the rest had their Scheme Managers paid by the promoters. In addition, the promoters were covering utility bills, equipment, and office and rental space among others. The promoters' subsidy costs are either from donor funding or facility revenues. The percent deducted from the premiums for administrative costs was generally used to procure scheme identify cards, take pictures of beneficiaries, enrolment costs, and leaders transport costs among other minor expenditures.

**The practice of strategic purchasing.** Strategic purchasing was assessed in terms of whether benefit packages were well defined, the existence of gatekeeping, and the prior agreement to the pricing of services. The schemes had a Memoranda of Understanding (MOU) that stipulated the roles and responsibilities of each party in managing, running, provision paying for healthcare and the risks involved. Except in one facility (located in Mubende district) where the CBHI is paying flat fees, the rest of the MOUs don't pronounce themselves on the pricing of the service. The community and third payment-managed CBHIS presented a clear definition of roles concerning purchasing and service provision. One of the scheme managers had this to say about the MoUs:

*"It is the service provider that prepares the package in the MOU and thereafter it is discussed by all parties" (Scheme Manager-KI 5)*

With exception of the provider-managed schemes that keep the scheme money, the others pay for service provision every month. The payments would be made after the issuance of bills by the facility, verified by the scheme managers, and approved by scheme leaders or fund managers in the case of community and third-party management respectively. Cheques were normally issued, or money was directly deposited into the facility account. There was no scheme paying cash to the facility as stated by one of the respondents:

*"We don't give money or cash directly; we pay through the bank" (Scheme Leader/Beneficiary, KI 12).*

With respect to the benefits package, all three types of CBHIS had well-defined packages. On the other hand, health centers have CBHIS covered for all services provided at the facility.

Also, the provider-managed schemes had no gatekeeping. Gatekeeping is used in this study to refer to the principle of when a member of the scheme falls ill or needs to be referred to a higher level of care, he or she contacts the lower level or primary care level first, which in turn authorizes care at a higher level within the scheme network of providers. The scheme member needs a referral from the lower level before seeking care outside the primary care level. In the community and third-party schemes, gatekeeping was feasible. The gate-keeping mechanism was incorporated into the package design. On gatekeeping, one of the Scheme Managers narrated as follows:

*".....the beneficiaries choose which lower facilities they will go to first and they also choose the a higher-level facility and, in this case, it's a hospital. They cannot go to the hospital before they are referred. Otherwise, if they go without a referral, the higher-level facility charges them by increasing the amount of co-payment depending on the cost of care. Say for example if the bill was UGX 50,000 = (USD 13.51) and the co-payment is meant to be UGX 7,000 = (USD 1.9), the provider can choose to charge such a beneficiary around UGX 30,000 = (USD 8.1) for skipping the lower facility. It is working . . . . . . . . .the referral network is within our system of healthcare and is discussed with the scheme. It starts from facilities at the lower level of care to our hospital. A referral card is given to the member to help him access the services from one level to another". (Scheme Manager-KI 13).*

### Best practices by CBHIS in Uganda

This study identified several best practices by the CBHI schemes. First, we found out that members were likely to join schemes, which they could trust in terms of managing their funds. Across the schemes, it was noted that the continued transparency and proper accountability seemed to encourage more members to join. The regular financial reports to the beneficiaries were stated to increase their trust in the schemes and attracted more to join. In addition, the old members were retained. The trust was also built by involving the members in the management of the CBHIS.

Second, it was noted that many schemes generate demand by first sensitizing the target community about general health insurance. People would be told about the specific benefit packages when they show interest. The provision of health insurance education was noted to be a continuous process throughout the life of the schemes. However, health insurance education was reported to be a very big cost that could not be covered by the premium.

Third, the community-based schemes to a big extent improved the health-seeking behaviour of the members. The possession of a healthcare card encourages them to seek healthcare early and easily at the facilities without worrying about how to pay the bills or first look around for the money before they go and thus avoiding the consequences of late reporting.

### Respondents' views on the integration of CBHIS into NHIS

**Why was integration considered necessary?.** When asked about integration and whether they thought it was necessary, all respondents felt that the integration of CBHIS into NHIS was urgently needed. Most of the reasons pointed to the need to accelerate achieving universal health coverage for the majority of the population.

Some of the national and district-level key informants had this to say:

*"This country is striving to achieve universal health coverage, but it has failed with the free healthcare provision in public facilities. One way to quicken this is by integrating CBHIS in*

*the proposed NHIS so as to make a contribution to broadening the coverage" (National level, KI 6).*

*"There are funding gaps in the healthcare system. Yet, the CBHIS provide evidence that a certain population within the informal sector can pay for their healthcare. So, they should be integrated to tap into those resources. Their contribution though small will form additional funds to ameliorating the funding gaps in the health sector" (National level, KI 7).*

*"The CBHIS are improving access to healthcare for their beneficiaries. But the purpose of NHIS is social protection. So, the government should ensure that CBHIS are integrated into the NHIS to provide social protection to those people in the informal sector" (District level, KI).*

However, some views were against the full-scale integration. They felt that integration should be gradual. That is, it should come later after the NHIS has operated for some time as government improves the capacity of service delivery at health facilities.

Some respondents also thought that sharing of resources would be encouraged through cross-subsidization (both vertical and horizontal). Those who have few resources would benefit from those who have much. This would however be realized if one fund is maintained for both formal and CBHIS. On the side of the government, the integration would benefit from building on existing structures that have been around for over 20 years.

*"The CBHIS are already trusted, and government can be part of it instead of us coming from NHIS to try to establish from national to the district and lower level which does not enhance ownership. Yet if we work with CBHIS, the ownership sense will be maintained by the grassroot people. This will also be efficient on side of government- since there are certain structures that can be worked on to ensure that the the scheme work well" (Central level, KI 1).*

**How the integration of CBHIS into NHIS can put on the national agenda?.** The majority of the KIIs (65%) felt that this was an important national issue that could easily be mixed up by the politics of the day. Thus, it would be good to discuss the integration of CBHIS into NHIS after election time. At the time of data collection, Uganda was in the process of holding national and local government elections. One of the national level KI explained as follows:

*"No one including the members of parliament and the president will allow to discuss such issues at this critical time. No political leader will want to associate with such a discussion at this time. Keep it until after the elections in March 2021, and then bring it. All politicians from local to central level will support it then" (National Level, KI 1).*

However, a section of some respondents thought the timing was perfect to present such an issue to the politicians. They noted that the NHIS issue was sighted in the Manifesto of the current president during the three previous presidential election campaigns. Also, experiences from Ghana show that NHIS which includes CBHI was on the political manifesto of one party during the presidential candidate's campaign at the time. The winning party automatically embraced it as a way of fulfilling its pledge to its people.

*"The population can get this issue of integration. They should push for it now and that approach might try to help, after all, user fees abolition was a political pledge also" (National level, KI 1).*

Another KI at the National level noted that:

*"Even when people think it's not feasible to present such issues to politicians at this time, I believe it is. Something happens during this political period; sometimes it's the right period to bring something on the agenda and it gets passed. . . . . . . . so, if it's the agenda of the electorates why do you keep it after elections, yet people are asking, and the politicians are making promises of what to do".*

Also, a National level KI from the civil society supported the idea that integration should be pushed now, and this is how it should be done:

*". . . . if CBHIS work with Civil Society Organizations (CSOs) and they push the candidate to include CBHIS into the NHIS this might work".*

Several respondents emphasized the need to avail compelling evidence to policymakers about the importance of integrating CBHIS into the NHIS and how feasible it is.

## Respondents' opinions on pooling

The current Bill on the national health insurance scheme proposes a single pool of funds, where the rich will subsidize the treatment of the poor, the healthy will subsidize the treatment of the sick, and the young will subsidize the treatment of the elderly. The Bill provides for mandatory contributions for any person who has attained the age of 18 years and who is ordinarily a resident of Uganda.

In this study, there were conflicting views on whether the integration should be done within a single pool of funds. A section of respondents felt that community-based health insurance schemes could be integrated into NHIS but with a separate pool of funds. Some respondents suggested creating a separate Bill for CBHIS. They argued that NHIS Bill has already progressed; therefore, integration might delay the process. Others felt that since integration is a process as opposed to a single event, it could be done in a phased manner; where CBHI schemes would first remain autonomous for some time but regulated together with NHIS. Those in favour of integrating CBHIS within a single pool of funds argued that it would ease the cross-subsidization from the formal to the informal sector and do away with the fragmentation of health funds.

*"The best scenario is to have one fund which each of us draws from. The only thing that can be done is to differentiate services based may be on category but fragmenting the health insurance fund was refused (from the other bill). So, the best is to have one fund which everybody draws from. It is also easy for development partners to contribute to such a fund and say, I'm paying for HIV services or for this group other than having "one thousand funds" and also having "one thousand managers" (National level, KI 1).*

**Increasing social acceptability after integration of CBHIS into NHIS.**   Among the suggested ways to increase acceptability after the integration of CBHIS into NHIS was to design a specific and attractive benefits package for the formal sector to ensure quality services. The reason given was that this group is comprised of a young population that does not often fall sick, yet the group can contribute more. More suggestions included the need for government to continue investing in community sensitization for demand generation for health insurance and primary health care. Particularly, there were ideas to have well-packaged campaign

messages to reach out to the formal sector. It was noted that proper messaging is important to have the formal sector accept subsidizing the informal sector. Some of the respondents at the national level emphasized this point as follows:

*"……….. design a message that is easily acceptable to the formal sector. There is no need to tell them outwardly that your premiums will subsidize the informal sector. The emphasis should be on the benefits the formal sector will get yet at the same time you develop a mechanism of having a percentage of every premium paid by the formal sector population to subsidize the informal. The dynamics of how you calculate and subsidize each other should be behind the system but sensitize the formal sector on the benefits of the NHIS" (National level, KI 1).*

*"The other way is to conduct advocacy; a lot of it with the formal sector to make government and donors appreciate the need for cross-subsidization. Use the CSOs in health to help with this task. Also, sensitize the local communities to understand the purpose of the scheme and its benefits (National level, KI 2).*

**Suggestions on technical integration.**   Three levels of resources or structures for CBHIS that can be incorporated into the integration were identified and presented. The first level comprises the CBHI promoters. These are the civil society organizations or the private sector that implement or support community-based health insurance schemes. Examples are: Save for Health Uganda, Community Based Health Financing Association (UCBHFA), HealthPartners Uganda (HPU), Integrated Community Based Initiatives (ICOBI), Uganda Protestant Medical Bureau (UPMB), Uganda Catholic Medical Bureau, hospitals like the Church of Uganda (COU) Kisiizi Hospital, Bwindi among others. The second level is comprised of community-based health insurance schemes themselves with their governance and management structures. While the third level comprises the community grass-root structures including local people with the necessary skills and knowledge, who are engaged in member enrolment, mobilization, and sensitization for health insurance within the communities.

The respondents made suggestions on how these structures could be integrated and what roles they could play in the integration. For instance, different respondents suggested that CBHI promoters could be taken on board in the integration to continue supporting the existing schemes and scale up their coverage within and beyond the current districts. They could also intensify efforts for health insurance education and community sensitization.

It was established that different strategies are required for the different segments in the informal sector. That is, for those in known and traceable work (organized informal sector) and those who are not easily traceable through occupation (unorganized informal sector). However, important to note is that the CBHIS are currently working with both segments (i.e., the organized groups and those hard to trace through occupation, but often reached through households). The schemes rely on the grass-root people with hands-on skills in mobilization and premium collection to reach these groups. Given this situation, it was strongly suggested that the integration should consider this arrangement and ensure that NHIS works with the current CBHIS grass-root structures to mobilize and collect premiums. They could be named the premium collection agents as one of the key informants pointed out:

*"We cannot lose the CBHIS grass-root people who are knowledgeable. I'm thinking that the CBHIS would come with their beneficiaries and say these are the members. So, maybe we would enter an MOU or whichever arrangement and get them to be part of the scheme. So, we can then contract their scheme to do more work and their leaders and scheme managers*

*would then be at the forefront of doing the work upon which we could have agreed upon. I know if the integration happens, then there will be an increased workload, so the current one will be observed but they will not even be enough. We will need more even as the enrolment increases" (Central level KI 1).*

**Challenges to integration.** When asked about the challenges that could limit the integration, different respondents pointed to the quality of healthcare in both public and private facilities. There were concerns that it would be difficult to satisfy the big number of contributors (from the formal and informal sectors). Further, the capacity of most facilities to manage big numbers of insured patients was reported to be inadequate. To address this, a result-based financing approach to health facilities was proposed.

One of the key informants at the national level responded as follows:

*"Our facilities are not able to handle big numbers since there are staff gaps and infrastructural challenges. This is worse when you get into the countryside, as a lot of regional imbalances exist. Yet the insured expects to receive quality services. Ministry of Health needs to first work on this issue. . . . . . ." (National level, KI 2).*

It was also reported that integration may be limited by the attitude of service providers and the target consumers themselves. Respondents noted that the commercial health insurance system has distorted the public perception of health insurance. Insurance companies have been delaying paying while peddling a lot of excuses.

Another reported challenge was that CBHIS were few and fragmented with some having very limited capacity and coverage. They also lacked standard guidelines. Moreover, their national association (UCBHFA) was not functional:

*"I see challenges in agreeing on what can be integrated. Most of those CBHIS for sure they told you that for us we want to remain the way we are running, alongside the NHIS, let them give us money or let us remain there and compete or if we are to be integrated, then we need a,b, c. . . . giving unfeasible conditions" (National level, KI-6).*

*"CBHIS cannot be integrated with their current numbers. Otherwise, taking on the current CBHIS (which has less than 150,000 people) will not make sense at all. If they are to be integrated, then they should be supported to enrol more people in their catchment area to be part of CBHIS. However, the CBHIS are not paying an actual premium, and therefore, what they pay cannot cover them for a full benefits package. Besides, some of the people within their communities are left out because they cannot pay. . . . . . . This might not be an easy task though and it's not clear if the government has the resources" (National Level, KI-7).*

**Options on stakeholders and the roles they can play in the integration.** Respondents gave options on who could be involved in the process of integrating CBHIS into NHIS and what roles they can play. The details of the suggested stakeholders and their roles are in S7 Table.

**Respondents' options on lessons learned.** The study documented several lessons learned by CBHIS implementers. First, people join and stay in a CBHI if they are sure of getting quality services. Satisfaction of beneficiaries by offering them quality healthcare is very important in minimizing dropout from CBHI. As such, poor quality of health care is a driver for members to drop out. Even those that might fear joining are enticed to be part of the CBHI. Second,

even with low premiums, some families (the poorest of the poor) may still fail to register with the scheme despite their willingness. Third, long distances and high costs of transport to healthcare providers undermine enrolment. Some people were not joining CBHIS whose service providers were perceived to be far especially if they had alternatives. Fourth, some families fail to pay the premiums when they have conflicting interests, especially in school fees payments and festival seasons coinciding with the premium payments period. Whenever CBHI renewal overlaps with school fees paying periods, it affects retention in CBHIS. Normally CBHIS have a grace period of 1 week to 1 month after opening a new year to ensure that such families also pay. Also, fluctuations in incomes and unstable incomes affect renewal because most of the beneficiaries depend on farming. So, whenever there is a poor harvest, they cannot afford the premium and end up dropping out. Fifth, the low insurance literate level of members makes it nearly impossible for them to run the schemes themselves. This is very common in community-managed CBHIS. This implies community members alone may not fully manage the technical aspects of the entire CBHIS.

## Discussion

This is the first study examining the possibility and providing pathways to integrating CBHIS into the proposed NHIS in Uganda. Ghana and Rwanda provided good examples in sub-Saharan Africa for integrating CBHIS into the NHIS. In both countries, the integration was done in a phased manner [11, 13, 14, 20–22]. In Ghana, the new Act of Parliament led to i) suspension or merger of the individual MHOs and their networks, and ii) consolidation of the DMHIS into the NHIS under NHIA management. The NHIA licenses and regulates the DMHIS [20–22]. Rwanda has over 500 pools at the community level, 30 district pools, and 1 national pool with only one national-level coordination mechanism. Responsibilities have been separated whereby the Ministry of Health is responsible for policy formulation and service provision at health facilities and hospitals and RSSB is charged with policy implementation and purchasing [11, 12, 14].

The functionality of the CBHIS in Uganda: The state of CBHIS in Uganda was to that in Ghana and Rwanda prior to integration. In both Ghana and Rwanda, the CBHIS were initially fragmented and autonomously managed with very limited coverage. For instance, in Ghana even with support from development partners like Danish International Development Agency (DANIDA), and the United States Agency for International Development (USAID), by the year 2000, the country had approximately 140 CBHIS covering less than 2% of the Ghanaians. More evidence showed that without integration into a single pool, fragmented CBHIS remain small with inadequate coverage, and this limits efforts to provide financial protection for all [8, 18, 22, 31–33]. At 7.6%, the penetration rate of the CBHIS in Uganda is relatively high compared with findings elsewhere. Scholars have suggested that CBHIS' penetration rate normally ranges between 3% to 5% and further state that it rarely reaches 10% [33–36]. Thus, this evidence indicates that CBHIS in Uganda could have a better penetration rate than in other countries. The affordability of the premium is one factor that affects the penetration rate of the CBHIS in Uganda. This finding is similar to studies done in many other low-income countries including Nigeria, China, India, and Burkina Faso [37–44]. Nevertheless, affordability was one major factor considered by the CBHI promoters at the time of determining and costing the benefits package. The practice of different premiums paid by beneficiaries in fragmented CBHIS is not new. It was also evident in Ghana, Tanzania, and Rwanda [11–30]. The Ugandan CBHIS implements co-payments to control abuse of the healthcare fund. Co-payment exists in the Rwanda CBHIS even after integration [13, 14]. However, the mode of co-payments observed in Rwanda differs from the Ugandan scenario. Whereas co-payment in Rwanda is

collected and used for administration costs, co-payment in Uganda CBHIS reduces the bill of the patient to the scheme. Premium payment in CBHIS had to be targeted in specific periods that were more convenient to communities. Unlike Ugandan CBHIS where some schemes pay on a quarterly or semester basis, CBHIS from other countries even before the integration paid an annual premium [11–14, 18–30]. In Rwanda, installment payments by households were noted [11–14] just like the Ugandan CBHIS. In Tanzania, most of the people are substance farmers and suggested that premiums could be collected after harvest. Thus, the premium collection period was out of season for the contributors [26].

Financial sustainability of the CBHIS: Evidence about the financial sustainability of the CBHIS in Uganda demonstrated that, members' active involvement in managing CBHIS improved trust concerning financial management. This finding is similar to what other scholars have documented [45]. In Ghana, members' involvement in the decision-making and management of CBHIS was found to have a significant contribution to the success of such a scheme [20–22]. This could have influenced the initial design of Ghana's NHIS [22, 23] which saw the adoption of organization and governance features of CBHIS integrated into NHIS. Similarly, in Rwanda, the structure and management of CBHIS were built at a decentralized level of districts [11–14]. Scholars [9, 22, 32, 33] put CBHIS networks as platforms for facilitating policy advocacy, technical support, and re-insurance among others. Also, the Senegalese government [32, 33] got involved in supporting the CBHIS to form such networks. In this study, some key informants pointed out that the Ugandan government's support CBHIS network was in its initial stages but later diminished. In fact, most respondents desired to see such support reinstated. Furthermore, the practice of donor funding in CBHIS is not new even in countries that have been successful in integrating CBHIS into NHIS. Several development partners like USAID, and DANIDA among others have previously planned a big role in ensuring the expansion of CBHIS before integration. This was evident in both Ghana and Rwanda [11, 13, 14, 20–23]. Some scholars [46, 47] documented that subsidizing CBHIS was a feasible approach if such schemes are to be sustainable in meeting the healthcare needs of its population. Such subsidies are not only put to the management of CBHIS but also extended to paying premiums for the poor. The subsidies are normally from government and development partners. Like the other CBHIS in Ghana, Tanzania, and Rwanda [11–14, 18–30], CBHIS in Uganda use the fee-for-service method of provider payment which some scholars paint as a less preferred method compared to the numerous alternatives available. Even with the integration, the district schemes in Rwanda still use fee-for-services [13, 14]. It was however noted that some CBHIS in Ghana used cash payment methods to pay providers. This provider payment method was mostly used before integration [23, 24]. Nevertheless, the CBHIS in Ghana practiced a mixture of provider payment methods and never relied on one as displayed in the Ugandan CBHIS. Contrary to Normand and Weber [48] recommendation of operating reserve size determined by the magnitude of the health fund, most of the reserve funds by the provider-managed schemes in Uganda were found to be empty. That is, in larger health funds the risks are widely distributed, and smaller health funds require a proportionally larger reserve. Normand and Weber further recommend that the reserve fund should equal at least two months' revenue of the scheme [48].

Integration of CBHIS in the NHIS: Documented respondents' views on integration showed that political leadership and commitment were important in enabling integration. In particular, resilient government support was deemed critical for ensuring the necessary legal framework, regulation, financing, subsidizing CBHIS, and fraud protection among others. In Tanzania, the idea of integrating CBHIS was mainly advanced by development partners with a limited political will; in the end, it was rejected for lack of proper design [27–30]. This was an indicator that integration was not among the priorities of the government. On a rather

different note, in Ghana, CBHI was a top priority on the president's agenda during and after the election and a deadline of fewer than 4 years was given to those that were designing it [21]. A similar view was held for Rwanda as scholars [11, 13, 14] indicate that the government's leadership and commitment directed the CBHI policy. Concerning their views on pooling, there was no consensus about a single pool. In Rwanda and Ghana where the integration of CBHIS into NHIS happened, it was noted that fragmented pools were consolidated into one pool. Even though this pooling is done in Rwanda at three levels, evidence of cross-subsidization, which is the major purpose of pooling, is visible. Having a single pool is advantageous in terms of having several sources of funds [13, 14, 27, 29, 46, 47]. The suggestion on technical integration indicated that three levels of the CBHIS would be incorporated into the NHIS. However, integrating CBHIS into NHIS requires specialized technical work to ensure financial visibility and technical commitment [46, 47]. Technical experts design the integration to cover a large segment of the population and safeguard against the inherent problems of CBHIS. On the same note, the technical experts also do respond to political urgency and realities. It is important to realize the technical and political objectives. In Ghana, it was noted that during the initial stages of integration, the DMHIS was governed by locally elected boards and managers following the spirit of CBHIS, but this was later disbanded in 2008 [20–24]. A list of the different stakeholders and the role they would play in the integration was suggested. Like Rwanda and Ghana, different stakeholders played different roles during and after the integration of CBHIS into NHIS. These roles ranged from technical, financial, advisory, and research among others [11–14, 20–24].

## Limitations of the study

The study was conducted during the COVID-19 pandemic. This caused several challenges, and the key was a limitation of mobility to some study sites. However, the study team used data abstraction based on the records presented, and many scheduled face-to-face interviews were replaced by telephone interviews. Interviews were carried over serval rounds to seek clarification, and scanned copies of data were requested from the study sites for verification. The study team found that some schemes have previously been operating as community-managed but by the study time, they had changed into provider-managed. The study team had to redo the sampling to have all three types of schemes interrogated. The search was limited to Google scholar and published articles in English. This may not have been comprehensive enough to cover enough literature on the three country case studies. However, checking the reference lists of the articles and other documents found expanded the list of the included literature.

## Conclusions

Our study demonstrates that it is possible to integrate the schemes into the National Health Insurance Scheme. In particular, the CBHI can be integrated into NHIS rather than being absorbed to create leverage for the schemes to be part of the national scheme and create conditions acceptable to all stakeholders. This can be done in a phased process. Out of this study, we make several recommendations:

   (i)The Health Sector could start by consolidating the existing 28 schemes at the district level into bigger District Health Insurance Schemes. The districts could pass ordinances making the CBHIS mandatory, but the implementation (enrolment) be done gradually. The compulsory nature of the schemes would stimulate universal enrolment in the district schemes. Once the district schemes are strong enough and have good coverage, then they can be integrated into the NHIS. The DHIS can be supported with government subsidies to cater for the indigents, which will in effect improve enrolment and pooling at the district level. Consolidating the

district-level schemes is critical for ensuring decentralized management to reduce structural inefficiencies and to cater to referrals.

(ii) The health sector and in particular MOH, the NHIS, and other promoters could integrate CBHIS in a phased manner into a larger consolidated district scheme given that they are fragmented with limited coverage and autonomously managed in three different ways. A district scheme will be near enough to the beneficiaries to have control and appropriately influence scheme operations. This will allow adequate integration of the existing structures, which would promote community participation and buy-in. Given the disjointed nature of the existing schemes without harmonized standards, it would be difficult to absorb them at once into a single NHIS.

iii) The Ministry of health could source technical assistance from NHIS and other promoters should then be provided to schemes at the district level to address the critical capacity gaps in CBHI. Government should provide this technical assistance through the proposed Board responsible for the supervision of NHIS. Since the district-level schemes will be a consolidation of the fragmented CBHIS, the technical assistance would help to address the critical capacity gaps in CBHIS identified by this study. Technical capacity should focus on creating standard pooling and purchasing practices to consolidate DHIS before integration into the NHIS. The technical assistance could also cover clinical care and financial audit of the schemes.

iv) Once MOH Uganda goes ahead with integration, it should include all three levels of community-based health insurance structures: CBHI promoters, the schemes, and the community grass-root structures. The CBHI promoters can be engaged at the district level, to continue popularising and supporting the existing schemes to scale up their coverage within the districts. They need to be supported by the government to intensify efforts for health insurance education and community sensitization. The capacity of individual schemes needs to be strengthened through technical assistance to feed into the consolidated DHIS. The grass-roots community structures that are comprised of resourceful persons with hands-on skills and knowledge in mobilization and premium collection should continue to reach out to the informal sector. These groups should be supported as community champions for community health insurance.

v) The Government of Uganda could consider one fund for both the formal and informal sectors managed at the national level as the final phase of integration. This recommendation is in line with the current Bill on NHIS that proposes a single pool of funds, where the rich will subsidize the treatment of the poor, the healthy will subsidize the treatment of the sick, and the young will subsidize the treatment of the elderly. Also, the fund would have a decentralized pooling at (i) the national level (i.e. contributions from salaries of formal sector workers, percent of district-level pooling, government and development partners); (ii) district level (from national pooling and CBHI schemes and development partners); and (iii) CBHI schemes (from members' contributions, partial or full subsidies for the poor and other vulnerable people from Government and Development Partners).

vi) A standalone option: A top-down consolidation strategy making full CBHI scale-up possible by having a national legal framework that recognizes CBHI and where all Districts or Local Governments are by law required to establish CBHI under the proposed NHIS. This approach shall require strong stewardship, legislation, and political commitment starting with districts with existing schemes and a rollout timetable to attain countrywide coverage. This has to be coupled with a clear strategy to improve the quality of healthcare.

## Supporting information

**S1 Table. Study districts, CBHIS types, and service providers.**
(DOCX)

**S2 Table. List of key informants.**
(DOCX)

**S3 Table. Experiences about the integration of CBHI into NHIS: Rwanda, United Republic of Tanzania, and Ghana.**
(DOCX)

**S4 Table. Coverage and current enrolment for CBHIS in Uganda.**
(DOCX)

**S5 Table. Example of different packages and ceilings from a third-party managed scheme.**
(DOCX)

**S6 Table. Functionality and financial sustainability aspects of CBHIS.**
(DOCX)

**S7 Table. Proposed stakeholders and roles.**
(DOCX)

**S8 Table. Capacity assessment tool of CBHI schemes in Uganda.**
(DOCX)

## Acknowledgments

Thomas Maine of Uganda Health Systems Supports Activity/USAID is acknowledged for his technical guidance during the validation meetings. We would also like to thank the key informants that provided the information needed.

The Ministry of Health of Uganda for allowing the authors to use the draft material: http://library.health.go.ug/publications/health-insurance/feasibility-analysis-community-based-health-insurance-scheme which contributed a substantial part to the development of the manuscript. Permission is attached.

## Author Contributions

**Conceptualization:** Prossy Kiddu Namyalo, Boniface Mutatina, Sarah Byakika, Aliyi Walimbwa, Robert K. Basaza.

**Data curation:** Prossy Kiddu Namyalo, Boniface Mutatina, Robert K. Basaza.

**Formal analysis:** Prossy Kiddu Namyalo, Boniface Mutatina, Robert K. Basaza.

**Investigation:** Prossy Kiddu Namyalo.

**Methodology:** Prossy Kiddu Namyalo, Boniface Mutatina, Robert K. Basaza.

**Project administration:** Rose Kato.

**Supervision:** Sarah Byakika, Aliyi Walimbwa, Rose Kato.

**Validation:** Sarah Byakika, Aliyi Walimbwa, Rose Kato.

**Writing – original draft:** Prossy Kiddu Namyalo.

**Writing – review & editing:** Prossy Kiddu Namyalo, Boniface Mutatina, Sarah Byakika, Aliyi Walimbwa, Rose Kato, Robert K. Basaza.

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
