## [Decision Letter · Decision Letter 0]

19 Dec 2022

PONE-D-22-25883THE FEASIBILITY ANALYSIS OF INTEGRATING COMMUNITY-BASED HEALTH INSURANCE SCHEMES INTO THE NATIONAL HEALTH INSURANCE SCHEME IN UGANDAPLOS ONE

Dear Dr. Namyalo,

Thank you for submitting your manuscript to PLOS ONE. After careful consideration, we feel that it has merit but does not fully meet PLOS ONE’s publication criteria as it currently stands. Therefore, we invite you to submit a revised version of the manuscript that addresses the points raised during the review process.

Two reviewers have evaluated your submission and have identified several aspects which require attention, including important clarifications regarding the study design and methods. Please respond carefully to all of the reviewers' comments when preparing your revision.

We look forward to receiving your revised manuscript.

Kind regards,

Jamie Males

Editorial Office

PLOS ONE

Journal Requirements:

2. "PLOS requires an ORCID iD for the corresponding author in Editorial Manager on papers submitted after December 6th, 2016. Please ensure that you have an ORCID iD and that it is validated in Editorial Manager. To do this, go to ‘Update my Information’ (in the upper left-hand corner of the main menu), and click on the Fetch/Validate link next to the ORCID field. This will take you to the ORCID site and allow you to create a new iD or authenticate a pre-existing iD in Editorial Manager. Please see the following video for instructions on linking an ORCID iD to your Editorial Manager account: " ext-link-type="uri" xlink:type="simple">https://www.youtube.com/watch?v=_xcclfuvtxQ"

4. Thank you for providing information regarding the report on the Ugandan Ministry of Health website from which your study was drafted. Please include your statement regarding this matter from the cover letter in the Methods section of the manuscript

Reviewers' comments:

Reviewer's Responses to Questions

**Comments to the Author**

1. Is the manuscript technically sound, and do the data support the conclusions?

Reviewer #1: Yes

Reviewer #2: Yes

2. Has the statistical analysis been performed appropriately and rigorously? 

Reviewer #1: Yes

Reviewer #2: N/A

3. Have the authors made all data underlying the findings in their manuscript fully available?

Reviewer #1: Yes

Reviewer #2: Yes

4. Is the manuscript presented in an intelligible fashion and written in standard English?

Reviewer #1: Yes

Reviewer #2: Yes

5. Review Comments to the Author

Reviewer #1: ABSTRACT

Background - I would suggest you change the statement " where the rich will subsidize the treatment of the poor and the healthy will subsidize the treatment of the sick, young and elderly" to "where the rich will subsidize the treatment of the poor, the healthy will subsidize the treatment of the sick, and young will subsidize the treatment of the elderly".

Method - This is a cross-sectional study with qualitative data collection method and review of documents. The study design stated here is questionable? Some of the data collection methods mentioned such as questionnaires, observations and archive were not performed or reported in the study.

Result - Give the USD equivalent of the UGX 75,215

Conclusion- How did the authors arrive at this conclusion? The result presented in the abstract did not show this.

TEXT

General comments - The write-up seems too lengthen. There are minor grammatical errors that need to be corrected. Abbreviations should be used in full the first time, before their subsequent use. The United States Dollar equivalent of the money stated within the text should be included.

Introduction - The authors gave a good background on the topic, showed the gap in knowledge and stated the aim of the study.

Methods

Study settings- How were the districts selected? What sampling method was used? Please give details of the sampling technique.

Data collection methods-

1. I would suggest the first statement in this subsection is moved to the study design

2. Who were the questionnaires administered to? How was the data from the questionnaire analyzed?

3. What was observed? where were the observations carried out?

4. What do you mean by archive as a method?

5. All I saw reading through the methods were Reviews of literature and other documents, Key Informant Interviews and Secondary data analysis

RESULT AND DISCUSSION

These were merged in the write up

I would suggest the authors separate the results from the discussion

Lessons learnt

ow did you come about some points such as "Another important lesson is that long distances and high costs of transport to healthcare providers undermine enrolment. Some people were not joining CBHIS whose service providers were perceived to be far especially if they had alternatives" and "Furthermore, we learned that some families fail to pay the premiums when they have conflicting interests, especially in school fees payments and festival seasons coinciding with premium payments. Whenever CBHI renewal periods overlap with school fee periods, it affects retention in CBHIs". The result section did not mention these points as findings of the study but were raised as part of lessons learns

CONCLUSION

Paragraph 6, line 4 - I would suggest you change the statement " where the rich will subsidize the treatment of the poor and the healthy will subsidize the treatment of the sick, young and elderly" to "where the rich will subsidize the treatment of the poor, the healthy will subsidize the treatment of the sick, and young will subsidize the treatment of the elderly".

Other comments are within the manuscript

Reviewer #2: The paper provides a comprehensive account of Uganda's existing community-based health insurance schemes

(CBHIS). It also examines the feasibility of integrating the CBHIS into a national health insurance scheme. I enjoyed reading the paper; very insightful. I particularly like how the authors relate Uganda's case to other sub-Saharan African countries (Ghana, Rwanda Tanzania). I have a few minor comments, notwithstanding.

The first relates to the use of abbreviations. Please check all for consistency (e.g., should readers assume CBHIS, CBHI, CBHIs, CHBIS CHBIs are used interchangeably? Also see DMHIS DMHIs). Ensure all abbreviations are defined upon first use (see CHI, SACCOs, CBO).

Second, some sentences need to be checked. e.g.,

i. "The effort towards the creation of a national scheme comes at a time when there already existing CBHIS." (are missing?)

ii. "Most of the reasons pointed to the need to accelerate achieving universal health coverage since the majority of the population." (Is that an incomplete sentence?)

Third, I found it hard understanding the use of e.g. "1000/=". Please feel free to ignore this comment if it is a standard approach which I simply do not know.

6. PLOS authors have the option to publish the peer review history of their article (what does this mean?). If published, this will include your full peer review and any attached files.

Reviewer #1: **Yes: **Tope Michael Ipinnimo

Reviewer #2: No

---

## [Author Response · Author response to Decision Letter 0]

18 Jan 2023

The response to reviewer and editor comments have been provided for in three files attached under “attached files”, i) Response to Reviewers', ii) 'Revised Manuscript with Track Changes, and iii) Manuscript.

---

## [Decision Letter · Decision Letter 1]

7 Feb 2023

PONE-D-22-25883R1THE FEASIBILITY ANALYSIS OF INTEGRATING COMMUNITY-BASED HEALTH INSURANCE SCHEMES INTO THE NATIONAL HEALTH INSURANCE SCHEME IN UGANDAPLOS ONE

Dear Dr. Namyalo,

Thank you for submitting your manuscript to PLOS ONE. After careful consideration, we feel that it has merit but does not fully meet PLOS ONE’s publication criteria as it currently stands. Therefore, we invite you to submit a revised version of the manuscript that addresses the points raised during the review process.

We look forward to receiving your revised manuscript.

Kind regards,

Gilbert Abotisem Abiiro, PhD

Academic Editor

PLOS ONE

Journal Requirements:

Additional Editor Comments:

1. Questionnaire is an instrument of collecting data but not a method of data collection. Replace questionnaire with the appropriate method of data collection in the abstract

2. All abbreviations used in the abstract should be written in full or defined within the abstract (eg: CHI and CBHI)

3. Please ensure that the manuscript is thoroughly proofread before resubmission

Reviewers' comments:

Reviewer's Responses to Questions

**Comments to the Author**

1. If the authors have adequately addressed your comments raised in a previous round of review and you feel that this manuscript is now acceptable for publication, you may indicate that here to bypass the “Comments to the Author” section, enter your conflict of interest statement in the “Confidential to Editor” section, and submit your "Accept" recommendation.

Reviewer #1: All comments have been addressed

Reviewer #2: (No Response)

2. Is the manuscript technically sound, and do the data support the conclusions?

Reviewer #1: Yes

Reviewer #2: Yes

3. Has the statistical analysis been performed appropriately and rigorously? 

Reviewer #1: Yes

Reviewer #2: N/A

4. Have the authors made all data underlying the findings in their manuscript fully available?

Reviewer #1: Yes

Reviewer #2: Yes

5. Is the manuscript presented in an intelligible fashion and written in standard English?

Reviewer #1: Yes

Reviewer #2: Yes

6. Review Comments to the Author

Reviewer #1: (No Response)

Reviewer #2: Many thanks for addressing my concerns. It appears the paper still has "CHBIS". Please define upon first use or correct if you mean CBHIS. (see "Integration of CHBIS...")

7. PLOS authors have the option to publish the peer review history of their article (what does this mean?). If published, this will include your full peer review and any attached files.

Reviewer #1: **Yes: **Ipinnimo T.M.

Reviewer #2: No

---

## [Author Response · Author response to Decision Letter 1]

6 Mar 2023

All comments raised have been addressed in four files attached under “Attach files). These are, a 'Response to Reviewers', 'Revised Manuscript with Track Changes', 'Manuscript' and Capacity assessment tool of CBHI schemes in Uganda

---

## [Editor Report · Decision Letter 2]

27 Mar 2023

THE FEASIBILITY ANALYSIS OF INTEGRATING COMMUNITY-BASED HEALTH INSURANCE SCHEMES INTO THE NATIONAL HEALTH INSURANCE SCHEME IN UGANDA

PONE-D-22-25883R2

Dear Prossy Kiddu Namyalo,

We’re pleased to inform you that your manuscript has been judged scientifically suitable for publication and will be formally accepted for publication once it meets all outstanding technical requirements.

Kind regards,

Gilbert Abotisem Abiiro, PhD

Academic Editor

PLOS ONE
---

## [Editor Report · Acceptance letter]

5 Apr 2023

PONE-D-22-25883R2 

The feasibility analysis of integrating Community-based Health Insurance Schemes into the National Health Insurance Scheme in Uganda 

Dear Dr. Namyalo:

I'm pleased to inform you that your manuscript has been deemed suitable for publication in PLOS ONE. Congratulations! Your manuscript is now with our production department. 

Kind regards, 

on behalf of

Dr. Gilbert Abotisem Abiiro 

Academic Editor

PLOS ONE